# Efficacy of Low-Level Laser Therapy in a Rabbit Model of Rhinosinusitis

**DOI:** 10.3390/ijms24010760

**Published:** 2023-01-01

**Authors:** Seok-Rae Park, Younghwan Han, Su Jeong Lee, Ki-Il Lee

**Affiliations:** 1Priority Research Center, Myunggok Medical Research Institute, Konyang University College of Medicine, Daejeon 35365, Republic of Korea; 2Department of Microbiology, Konyang University College of Medicine, Daejeon 35365, Republic of Korea; 3Department of Medical Device Convergence Center, Konyang University Hospital, Daejeon 35365, Republic of Korea; 4Department of Microbiology, School of Medicine, Cha University, Seongnam 13488, Republic of Korea; 5Myunggok Medical Research Institute, Konyang University College of Medicine, Daejeon 35365, Republic of Korea; 6Department of Otorhinolaryngology-Head and Neck Surgery, Konyang University College of Medicine, Daejeon 35365, Republic of Korea

**Keywords:** sinusitis, low-level laser therapy, experimental model, cytokines, inflammation

## Abstract

Little is known about alternative treatment options for rhinosinusitis (RS). We aimed to evaluate the efficacy of low-level laser therapy (LLLT) for RS in experimentally induced rabbit models of RS. A total of 18 rabbits were divided into four groups: a negative control group (*n* = 3), an RS group without treatment (*n* = 5, positive control group), an RS group with natural recovery (*n* = 5, natural recovery group), and an RS group with laser irradiation (*n* = 5, laser-treated group). Computed tomography and histopathological staining were performed for each group. mRNA and protein expression levels of local cytokines (IFN-γ, IL-17, and IL-5) were also measured. Tissue inflammation revealed a significant improvement in the laser-treated group compared with the RS and natural recovery groups (*p* < 0.01). In addition, sinus opacification in the CT scans and cytokine expression was reduced in the laser-treated group, though without statistical significance. LLLT could be an effective option for the management of RS concerning radiological, histological, and molecular parameters.

## 1. Introduction

Rhinosinusitis (RS) is one of the most prevalent inflammatory disorders among upper respiratory diseases characterized by nasal obstruction, post-nasal drip, rhinorrhea, and facial pain [1,2,3]. The standard treatment for RS includes appropriate medical therapy, such as saline irrigation, antibiotics, and topical/systemic steroids. Endoscopic sinus surgery has been performed to treat cases of refractory or recurrent RS despite an appropriate medical therapy. However, there is a paucity of information with regard to alternative management for RS.

Low-level laser therapy (LLLT) is a form of phototherapy that involves radiation using photon energy at low levels to alter biological activity without causing thermal injury [4]. Compared with high-level lasers, LLLT uses energy at low levels that are sufficient to promote tissue response, leading to tissue regeneration [5]. Therefore, LLLT is considered to be non-toxic and non-allergenic and has been used extensively for medical purposes, such as tissue repair, pain control, and treatment of musculoskeletal disorders [6,7,8,9]. LLLT has also been applied to the upper and lower airways [10,11]. Due to its anti-inflammatory effects, it can be assumed that LLLT may also be effective as an adjunctive treatment for RS.

However, there is still a lack of clarity about the effects of LLLT on RS. A few clinical studies on the efficacy of LLLT involving small numbers of patients with RS have been conducted [12,13]. Clinically, they suggest that LLLT is a possible alternative treatment for RS. However, the study findings were inadequate due to relatively small sample sizes. Moreover, they suggested the clinical efficacy preliminarily, without elucidating the underlying mechanism.

Animal models have been used for understanding the pathogenesis of RS and the possible treatment options [14]. Experimental models for RS mostly use mice [11,15,16] and occasionally rabbits [17]. Using rabbit models for RS has several advantages [18]. Firstly, anatomically, rabbit RS is comparable to human RS due to their having a relatively large maxillary sinus. Secondly, histologically, sufficient inflamed sino-nasal mucosa can be easily obtained for microscopic analyses. Finally, methodologically, the experimental models are well established and can be easily utilized using pieces of Merocel sponge (Metronic-Xomed, Jacksonville, FL, USA).

We attempted to clarify the effect of LLLT on the sino-nasal mucosa of rabbits with experimentally induced RS using pulse-type lasers of 650/830 nm wavelength. We aimed to verify whether LLLT could reduce acute inflammation using histological, radiological, and molecular biological analyses.

## 2. Results

The experimental groupings and protocol are described in the Materials and Methods section.

### 2.1. Radiological Changes

Computed tomography (CT) was performed to confirm inflammatory status radiologically in all groups.

On day 14, empyema volumes measured using a micro-CT scanner were 0.0 ± 0.0, 684.09 ± 156.25, 851.92 ± 260.76, and 831.42 ± 131.23 mm^3^ in the negative control, positive control, natural recovery, and laser-treated groups, respectively. Empyema volumes were significantly different in the experimentally induced RS groups (positive control, natural recovery, and laser-treated groups) compared with the negative control group (*p* < 0.05, negative control vs. positive control groups; *p* < 0.05, negative control vs. natural recovery groups; *p* < 0.05, negative control vs. laser-treated groups). On day 21, empyema volumes were 557.96 ± 240.60 and 437 ± 142.10 mm^3^ in the natural recovery and laser-treated groups, respectively. Empyema volumes in the laser-treated group were reduced compared with those in the natural recovery group (*p* = 0.678). Thus, better radiological improvement was achieved by LLLT compared with natural recovery (Figure 1).

### 2.2. Histopathological Differences

The epithelial thickness was significantly hypertrophied in the RS-induced groups (positive control and natural recovery groups), whereas it was reversed almost to the control thickness by LLLT (laser-treated group) (*p* < 0.001, negative control vs. the positive control groups; *p* < 0.001, negative control vs. natural recovery groups). Overall infiltrations of inflammatory cells were notably observed in the positive control and natural recovery groups when compared with the negative control group (*p* = 0.09, negative vs. positive control group; *p* < 0.001, negative vs. natural recovery groups). Interestingly, infiltration was significantly restored by LLLT (laser-treated group) compared with that of the natural recovery group (*p* = 0.002). Likewise, infiltrations of goblet cells, eosinophils, and mast cells were significantly observed in the positive control and natural recovery groups when compared with the negative control group (*p* < 0.001, negative control vs. positive control groups; *p* < 0.001, negative control vs. natural recovery groups), whereas cell infiltration was notably reversed in the laser-treated group compared with that in the natural recovery group (*p* < 0.001). Thus, histopathological analyses revealed that RS could be treated more effectively by LLLT (Figure 2, Appendix A).

### 2.3. Local Cytokine Levels

The efficacy of LLLT with respect to experimentally induced RS was confirmed by mRNA and protein levels.

The mRNA levels of the typical Th1 and Th17 cytokines, IFN-γ and IL-17, in the sino-nasal mucosa increased after RS induction (positive control group) compared to the negative control. Slightly reduced mRNA expression was observed in the natural recovery group compared to the positive control group. Interestingly, relative mRNA expressions tended to be restored in the laser-treated group, though without statistical significance (Figure 3A). Likewise, the Th2 cytokine IL-5 showed similar trends to those of IFN-γ and IL-17.

IFN-γ protein levels measured by Western blotting were notably increased in the positive control group when compared with those in the negative control group (Figure 3B). Protein levels of IFN-γ in the laser-treated group showed a notable decrease compared to the natural recovery group, but without statistical significance (*p* = 0.243). IL-5 protein level showed a similar outcome according to Western blotting analysis. On the other hand, the protein expression of IL-17 did not show significant changes among all groups.

## 3. Discussion

The present study revealed that LLLT significantly improved experimentally induced RS. Radiologically, experimentally induced RS was confirmed using a piece of surgical sponge. Histopathologically, inflammatory cell infiltrations were significantly observed in experimentally induced RS rabbits compared to the negative control group. Immunologically, inflammatory cytokine expressions were increased in the experimentally induced RS groups compared with the negative control group. These parameters were restored after surgical sponge removal. Furthermore, LLLT resulted in accelerated improvement compared with natural recovery. Taking all these results together, it can be suggested that LLLT has a therapeutic effect on RS.

Numerous articles have reported phototherapies, such as ultraviolet or far-infrared therapies, as novel therapeutic tools for the management of allergic rhinitis (AR) [19,20,21,22]. Among them, a few clinical studies on LLLT have been carried out on living patients with AR. In particular, numerous commercial laser devices have been used to treat AR. Recently, Jung et al. reported a randomized, double-blind, placebo-controlled trial to clarify the efficacy and safety of LLLT among 67 patients with AR [23]. They verified that LLLT could be an effective and safe treatment for AR with respect to the sino-nasal symptoms and improve quality of life by applying the commercially available device intra-nasally in patients with AR.

However, only a few studies have investigated the effect of LLLT in patients with RS. In a pilot study involving 15 adult patients with RS, Naghdi et al. [12] demonstrated that subjective improvements based on a visual analogue scale and total nasal symptom score were significantly achieved using infrared lasers. Their study was meaningful because the clinical trial was conducted with living patients, even though it had a small sample size and only subjective parameters were evaluated. Similarly, Krespi et al. [13] reported the bactericidal and wound healing effects of near-infrared illumination on 23 post-surgical CRS patients. They asserted that near-infrared illumination was subjectively and objectively beneficial in managing CRS, being safe, reproducible, and sustained, and that it did not appear to damage ciliary movement. In the present study, we verified the effect of LLLT on RS based on objective parameters assessed in vivo using a rabbit animal model.

In terms of laser irradiation methods, numerous lasers of different wavelengths, output powers, and gain mediums have been used [24]. For LLLT, red and near-infrared light with wavelengths in the range of 390–10,600 nm and output powers of up to 500 mW have been used for medical purposes [4]. In our study, a multi-wavelength and muti-power system (AlGaInP 670 nm (visible ray), 3 mW; GaAs 830 nm (infrared ray), 20 mW) was used, whereas an 830 nm Ga-Al-As laser was used in continuous-wave mode at a power output of 30 mW and an energy dose of 1 J in the aforementioned pilot study [13]. However, it is not yet known which irradiation method is optimal for RS or even AR treatment.

There are several concerns regarding the adverse effects of LLLT. At an experimental level, several studies have reported side effects and DNA injury by LLLT [25,26,27]; hence, there could be a concern regarding carcinogenic potential caused by DNA damage. However, the damage can be reversed [28]. It has been decisively reported that LLLT at therapeutic doses promotes the expression of DNA repair genes and induces changes in the gene expression profiles of irradiated cells [29,30]. Tam et al. [31] reported that the responses of cancer cells to LLLT vary and may differ considerably among oncological types and laser settings.

In addition, high wavelengths between 700 and 1400 nm can be harmful to the eyes. However, the laser wavelengths (670 nm, 3 mW and 830 nm, 20 mW) we used are not hazardous to the eye [23]. Potential LLLT-related adverse events include dermatological manifestations, such as mucosal dryness, allodynia, pruritus, erythema, and pigmentation [20].

Histologically, several studies have revealed that LLLT increases collagen fiber deposition, promotes fibroblast proliferation, and enhances microcirculation in local tissues, which results in virtuous physiological cycles [32,33,34]. Our histopathological outcomes based on epithelial thickness revealed similar results, showing that LLLT can significantly reduce tissue fibrosis based on epithelial thicknesses. Moreover, we demonstrated that LLLT significantly reduced goblet cell, eosinophil, and mast cell depositions in the sino-nasal mucosa. Based on these histological findings, it can be considered that LLLT is suitable for sino-nasal mucosa.

The exact mechanism of LLLT is not yet fully understood. It is assumed to be effective and multifactorial with respect to RS. The anti-inflammatory activity of LLLT has been explained via several animal models [10,11,16]. The photobiomodulation effect of LLLT can lead to modulation of cell metabolism and reduced inflammation [35]. A review concluded that LLLT can control the inflammatory response by reducing inflammatory cells, such as TNF-α, cyclooxygenase-2, prostaglandin E2, and IL-1β [36]. Pires et al. [37] reported that LLLT can induce an anti-inflammatory effect that regulates the transcription factors associated with cyclooxygenase-2 through the modulation of the expression of pro-inflammatory micro-ribonucleic acid. Aimbre et al. [38] noted that LLLT reduced neutrophil influx and the expression of IL-1β micro-ribonucleic acid. LLLT can inhibit nuclear factor kappa B (NF-κB), the key inflammatory transcription factor, and the related signaling pathways [39,40]. LLLT can also suppress other important mediators of inflammation, namely, macrophage-associated inflammatory proteins and pro-inflammatory cytokines [41]. Nambi et al. [42] demonstrated that LLLT can reduce TNF-α and MMP-13, whereas LLLT seems to be ineffective with respect to changing levels of IL-6. In addition, studies on the molecular and cellular mechanisms of LLLT suggest that photons are absorbed by the mitochondria, leading to increased ATP production and increased reactive oxygen species generation [5,43,44].

Meanwhile, conflicting results regarding the effect of LLLT on cell proliferation have been published [45,46,47]. Studies have tried to demonstrate the effect of LLLT on DNA repair mechanisms, but their results varied due to methodological discordances [6]. LLLT also reduces the expression of IFN-γ, IL-1β, and IL-17 [48]. In our study, the representative Th1 (IFN-γ) and Th17 (IL-17) cytokine expressions were increased in experimentally induced RS rabbits when compared with the negative control group. Notably, these cytokine expressions were ameliorated in LLLT-treated rabbits when compared with the natural recovery rabbits.

Although our rhinogenic model induced mainly bacteriological and neutrophilic inflammation [49,50], allergic markers, such as IL-5 expression, revealed similar results for Th1 and Th17 cytokine expression. Furthermore, increased serum total IgE levels in experimentally induced RS rabbits were restored in laser-treated rabbits (Appendix A). Thus, allergy and RS affect each other in the airway mucosa [51]. Moreover, perennial allergy plays a significant role in chronic and recurrent acute RS, according to clinical studies [52].

This experimental study, it should be noted, has some limitations. Firstly, the present experimentation was performed for a short course of RS induction and laser irradiation. Secondly, we analyzed data based on limited sample sizes per group. Thirdly, we evaluated efficacy only, not the adverse effects. Lastly, we considered only maxillary sinusitis and not the other sinuses. In the future, a large prospective study involving living patients must be conducted to overcome these limitations and validate our findings.

## 4. Materials and Methods

### 4.1. Groupings, Rabbit Model of RS, and Tissue Preparation

The Institutional Animal Care and Use Committee of Chemon was accredited by the Association for Assessment and Accreditation of Laboratory Animal Care (AAALAC International). The committee approved the animal experiments, which were designed and performed in compliance with the governmental and international guidelines on animal experimentation (20-B711).

Eighteen female New Zealand white rabbits (9 weeks of age, 2.1–3.8 kg) were used for the study. Before initiating RS, all rabbits were acclimatized to the animal facility with sufficient food and water for 1 week. The rabbits were randomly divided into four groups (*n* = 18, total): a negative control group (*n* = 3), a positive control group (*n* = 5), a natural recovery group (*n* = 5), and a laser-treated group (*n* = 5) (Figure 4).

Before the RS induction procedure was performed, 15 rabbits (positive control, natural recovery, and laser-treated groups) were anesthetized with an intramuscular injection of 50 mg/kg ketamine hydrochloride and 5 mg/kg xylazine hydrochloride. Then, an approximately 3 cm sized maxillary bone window was made using a surgical drill (Figure 5). A 3 × 5 × 25 mm piece of Merocel sponge (Medtronic-Xomed, Jacksonville, FL, USA) was inserted into one sino-nasal cavity, while the other cavity was left untreated as the control side. Two weeks later, the Merocel sponge was removed, and the rabbits were sacrificed. Meanwhile, in the natural recovery group, natural recovery was achieved over 1 week by reopening the sinus ostium after removing the Merocel sponge. In the laser-treated group, additional laser irradiation was performed for 1 week. None of the rabbits received any medications, such as antibiotics or steroids, before or during the procedure.

The rabbits were euthanized, and their heads were removed *en bloc* on days 0, 14, and 21. Eighteen upper snout samples containing the maxillary sinus were fixed on 10% neutral-buffered formalin for histopathological analysis.

### 4.2. Laser Device and Application

Additionally, rabbits in the laser-treated group were subjected to LLLT (multi-wavelength and multi-power (AlGaInP 670 nm with 3 mW, GaAs 830 nm with 20 mW), Optowell Co., Ltd., Jeonju, Republic of Korea; Figure 6) for 1 week. Laser irradiation was applied to the nasal cavity of the rabbits under anesthesia. Detailed parameters are shown in Table 1.

### 4.3. Radiological Evaluation Using Computed Tomography

Using a micro-CT scanner with a multispecies preclinical imaging system (PerkinElmer Inc., Waltham, MA, USA), the rabbits were examined to evaluate RS statuses. CT was performed without inserting the Merocel sponge at day 0 in the negative control group and after 2 weeks of RS induction in the positive control, natural recovery, and laser-treated groups. CT was performed to confirm RS induction in the positive control, natural recovery, and laser-treated groups after inserting the Merocel sponge for 2 weeks. CT was reperformed to compare radiological inflammation between the natural recovery and laser-treated groups. The coronal plane where the maxillary sinus is visible at the largest size was selected for CT. In the selected plane, empyema volume (mm^3^) was calculated quantitatively using an automated image analyzer (*i*Solution FL version 9.1, IMT *i*-solution Inc., Bernaby, BC, Canada).

### 4.4. Histopathological Analysis

Histological sections were individually prepared for all 18 samples. Each upper snout sample was decalcified in decalcifying solution (24.4% formic acid and 0.5 N sodium hydroxide) for 5 days (mixed decalcifying solution was exchanged once a day). Each upper snout sample was trimmed across with the right maxillary sinus as one part and then embedded in paraffin using an automated embedding center (Shandon Histostar, Thermo Fisher Scientific, Waltham, MA, USA) individually and sectioned as four 3–4 μm thick serial sections using an automated microtome (RM2255, Leica Biosystems, Nussloch, Germany). Representative sections were stained with hematoxylin and eosin (HE) for overall inflammation, periodic acid–Schiff (PAS) for mucus-producing cells, congo red (CR) for eosinophils, and toluidine blue (TB) for mast cells. Mean epithelial thicknesses (μm) and numbers of inflammatory cells that infiltrated the maxillary sinus mucosa (cells/mm^2^) were calculated using HE staining and the mean numbers of infiltrated mucus-producing cells, eosinophils, and mast cells estimated using PAS staining, CR staining, and TB staining, respectively, under a light microscope (Model 80i, Nikon, Tokyo, Japan) equipped with camera systems (ProgResTM C5, Jenoptik Optical Systems GmbH, Jena, Germany) and a computer-assisted automated image analyzer (*i*Solution FL version 9.1, IMT *i*-solution Inc., Bernaby, BC, Canada). Histomorphometrical index estimation was performed by researchers blinded to the identity of the samples. One part in each upper snout (right maxillary sinus tissue), one histological field in each part of the maxillary sinus, and eighteen total snout samples were statistically analyzed in the histopathological inspection.

### 4.5. RNA Isolation and qPCR in the Sino-Nasal Mucosa

RNA was obtained from the sino-nasal mucosa for the assessment of cytokine gene expression by real-time PCR. Total RNA was extracted using an easy-spin^TM^ total RNA extraction kit (iNtRON Biotechnology, 17221). Subsequently, cDNA was synthesized using RevertAid First Strand cDNA Synthesis (Thermo Fisher Scientific, #K1622). Amplification of 1 µg of cDNA was carried out using the SYBR Green Supermix (BioRad Laboratories, Hercules, CA). The PCR program was performed on a CFX connector (Bio-Rad) as follows: initial pre-denaturation step at 95 °C for 3 min, followed by denaturation at 95 °C for 30 s, annealing and extension at 60 °C for 30 s. Relative gene expression was quantified using the comparative Ct method (2ΔΔCt), as described by the manufacturer. Data were normalized to rabbit *GAPDH* (glyceraldehyde 3-phosphate dehydrogenase) mRNA levels.

### 4.6. Western Blotting of the Sino-Nasal Mucosa Proteins

Sino-nasal tissues (90 mg) were lysed with RIPA buffer (Thermo Fisher Scientific) containing a protease inhibitor cocktail (Cell Signaling Technology, Danvers, MA, USA). Protein concentrations were determined using a bicinchoninic acid protein assay kit (Thermo Fisher Scientific). Total protein was separated using a 12% SDS PAGE gel (Thermo Fisher Scientific) and electrotransferred to polyvinylidene difluoride membranes (Thermo Fisher Scientific) using an iBlot2 Transfer Device (Thermo Fisher Scientific). The membranes were blocked with 5% skim milk (BD Biosciences, San Jose, CA, USA) at room temperature for 1 h and then incubated with specific primary antibodies, including monoclonal mouse anti-rabbit IFN-γ antibody (LSBio, LS-C292463-200, Seattle, WA, USA), monoclonal mouse anti-rabbit IL-17 antibody (LSBio, LS-C314130-200), and rabbit IL-5 polyclonal antibody (MYBioSource, MBS3211621, San Diego, CA, USA), and overnight at 4 °C. The membranes were washed with tris-buffered saline containing 0.1% Tween-20 and then probed with HRP-conjugated secondary antibodies, including goat anti-rabbit IgG H&L (HRP) (Abcam, ab97051, Cambridge, UK) and rabbit anti-mouse IgG H&L (HRP) (Abcam, ab97046). Bands were detected using SuperSignal West Femto Chemiluminescent Substrate (Thermo Fisher Scientific) and the LuminoGraph2 image system (ATTO). Samples from all four groups were analyzed for IFN-γ, IL-17, and IL-5 expression. Image J software (U.S. National Institutes of Health, Bethesda, MD, USA) was used for protein quantifications.

### 4.7. Statistical Analysis

The results for empyema volumes, cytokine levels, and sino-nasal histopathological analyses were expressed as the means ± standard errors of the means (SEMs), and cross-group comparisons were made using the Mann–Whitney U test. Comparisons between groups were performed using the Kruskal–Wallis test with Dunn’s multiple comparisons test. *p*-values < 0.05 were considered significant. The Mann–Whitney U and Kruskal–Wallis tests were implemented using Graphpad version 5.0 (Graphpad Software, San Diego, CA, USA; www.graphpad.com).

## 5. Conclusions

We demonstrated a significant improvement in RS by LLLT in a rabbit model compared with untreated and naturally treated groups based on radiological findings, inflammatory cytokine levels, and histopathological findings. From a clinical point of view, our results suggest that LLLT may be an alternative therapeutic option for RS.

## Figures and Tables

**Figure 1 ijms-24-00760-f001:**
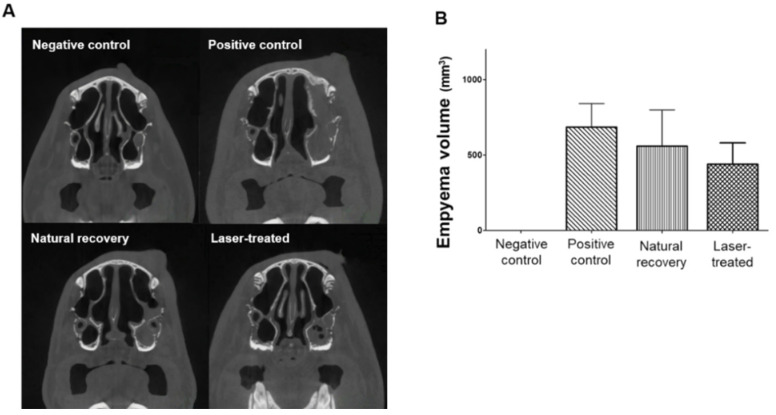
Representative images of and quantitative data for empyema volumes, estimated using computed tomography. (**A**) The negative control group showed no empyema in the maxillary sinus, whereas the positive control group showed total soft tissue density in the left maxillary sinus. (**B**) In the laser-treated group, empyema volumes were decreased compared with those in the natural recovery group. Data are expressed as the means ± SEMs. The results were statistically analyzed by Student’s *t*-tests.

**Figure 2 ijms-24-00760-f002:**
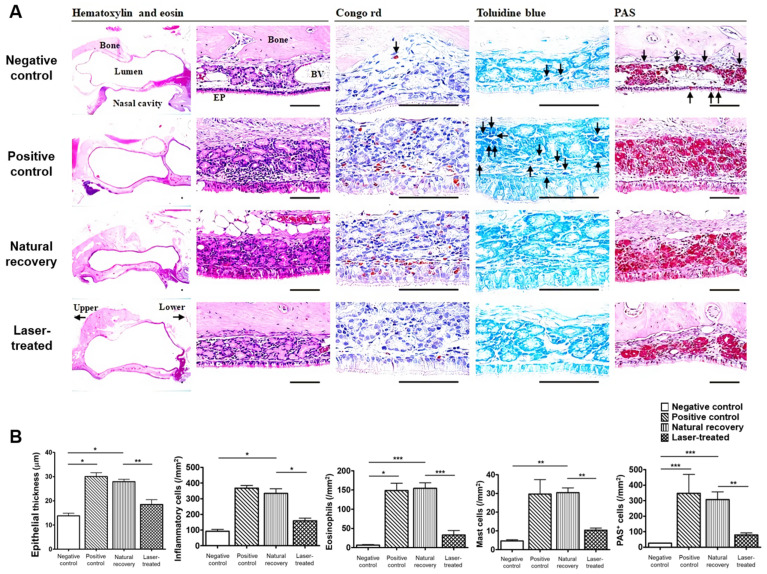
Representative histopathological images of the sino-nasal mucosa according to groups (**A**). Quantitative data of histopathology in the sino-nasal mucosa among all groups (**B**). Data are expressed as the means ± SEMs. * *p* < 0.05, ** *p* < 0.01, *** *p* < 0.001. Scale bars = 80 μm. Upper and lower means head to oral cavity direction. (Upper indicated head direction whereas lower indicated oral cavity direction) Arrows and dot colors indicate eosinophils in Congo red stains, Mast cells in Toluidine blue stains and goblet cells in PAS stains.

**Figure 3 ijms-24-00760-f003:**
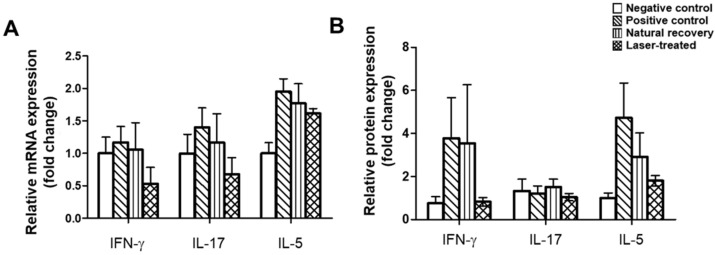
Local cytokine levels as determined by real-time PCR and Western blotting. (**A**) Sino-nasal mucosa RNAs were isolated and the levels of IFN-γ, IL-17, and IL-5 were measured by RT-qPCR. (**B**) Sino-nasal mucosa proteins were isolated and the levels of IFN-γ, IL-17, and IL-5 were measured by Western blotting. Graphs indicate relative protein expression levels normalized to β-actin using the densitometric analysis with Western blotting bands presented in Appendix A. Data are expressed as the means ± SEMs.

**Figure 4 ijms-24-00760-f004:**
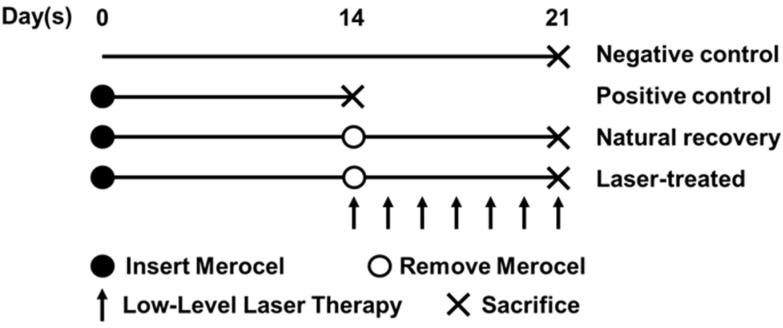
Experimental protocol. After 1 week of acclimatization, rabbits were divided into 4 groups: negative control group (*n* = 3), positive control group (*n* = 5), natural recovery group (*n* = 5), and laser-treated group (*n* = 5). Rabbits that were not subjected to experimental treatment (natural recovery group) and rabbits subjected to laser irradiation for one week (laser-treated group) were sacrificed on day 21.

**Figure 5 ijms-24-00760-f005:**
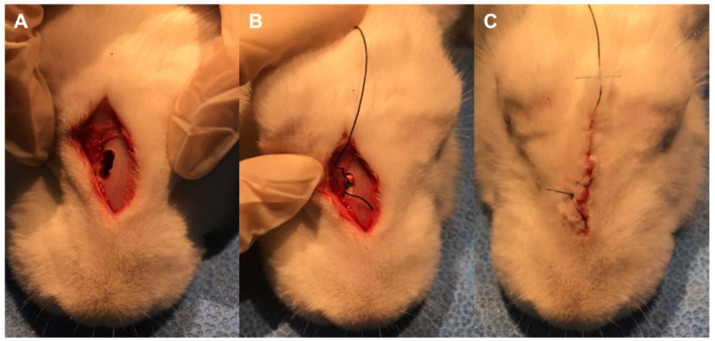
Induction of rhinosinusitis in a rhinogenic rabbit model. (**A**) After anesthetization, a 3 cm sized bone window was created in the anterior wall of the maxilla. (**B**) A piece of surgical sponge, Merocel (Medtronic-Xomed, Jacksonville, FL, USA), was inserted into the sino-nasal cavity to block the natural ostium of the sinuses. (**C**) The surgical window was closed using surgical thread. RS induction by inserting Merocel was performed for 2 weeks in the positive control, natural recovery, and laser-treated groups.

**Figure 6 ijms-24-00760-f006:**
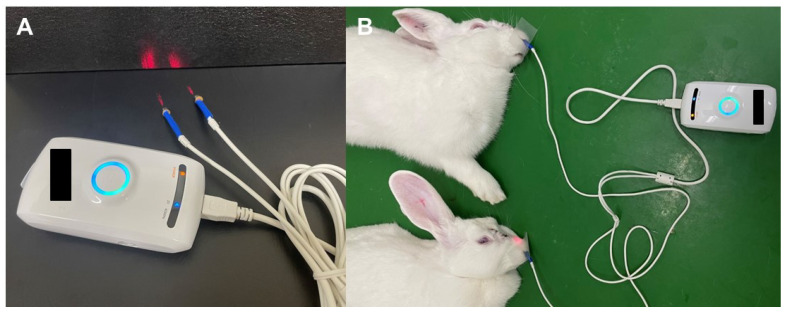
Laser irradiation of rabbits. (**A**) A low-level laser device was used at the following wavelengths and powers: AlGaInP 670 nm, 3 mW and GaAs 830 nm, 20 mW. (**B**) Low-level laser therapy was applied to five rabbits (G4) through the nostrils for 20 min for 7 days.

**Table 1 ijms-24-00760-t001:** Laser specifications used in the present study.

Variable	Number
Wavelength	AlGaInP 670 nm and GaAs 830 nm (multiple)
Output power	670 nm, 3 mW; 830 nm, 20 mW (multiple)
Total power	23 mW
Usage time	20 min

## Data Availability

Data are available on request from the corresponding author.

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
