# Peer review of "Efficacy of Low-Level Laser Therapy in a Rabbit Model of Rhinosinusitis"

_ijms, 2023, doi:10.3390/ijms24010760_

Round 1
Reviewer 1 Report
I found this study to be interesting and contain important results.
However, I thought there were several areas for improvement.
1. I think that the inclusion of pictures of animals requires a statement of ethical considerations. If there is an ethics committee for the experiment, please describe it.
2. Are you saying that there was no statistically significant difference in the results shown in Figure 3A and B ? I took the statement in line 113-117 as a statement that there is a significant difference.Please clarify wording. Any significant differences should be indicated by symbols in the figure.
Author Response
Point 1: I think that the inclusion of pictures of animals requires a statement of ethical considerations. If there is an ethics committee for the experiment, please describe it.
Response 1: Thank you for the observation. Absolutely, this experiment was approved by the ethical committee. And, we already described the statement for the ethical consideration in the materials and methods section as seen below.
The Institutional Animal Care and Use Committee of Chemon approved the animal experiments, which were designed and performed in compliance with the governmental and international guidelines on animal experimentation (20-B711).
We confirm that this committee which approved this experiment was accredited by AAALAC International. AAALAC International is an authorized, nonprofit organization that promotes the humane treatment of animals in science through voluntary accreditation and assessment programs. We added this description in the manuscript as seen below.
The Institutional Animal Care and Use Committee of Chemon was accredited by the Association for Assessment and Accreditation of Laboratory Animal Care (AAALAC International). The committee approved the animal experiments, which were designed and performed in compliance with the governmental and international guidelines on animal experimentation (20-B711).
In addition, please note the attached image (yellow mark) which confirms our committee was included in AAALAC accreditation.
Point 2: Are you saying that there was no statistically significant difference in the results shown in Figure 3A and B ? I took the statement in line 113-117 as a statement that there is a significant difference.Please clarify wording. Any significant differences should be indicated by symbols in the figure.
Response 2: We appreciate your comment. As the reviewer mentioned, the graph seems to show a significant difference among the groups. However, due to the lack of sample size and some outliers, there was no significant difference between groups. We confirm that no symbols indicating statistical significance in the graph is correct. We revised the wording properly in the manuscript as seen below.
Slightly reduced mRNA expression was observed in the natural recovery group compared to that in the positive control group. Interestingly, the relative mRNA expressions tended to be restored in the laser-treated group, however without a statistical significance (Figure 3A).

Reviewer 2 Report
The authors state that they have demonstrated the usefulness of LLLT in sinusitis. However, the research methodology of this paper needs to be reconsidered.
When comparing the laser treated group to the natural recovery group, all procedures must be the same except for the laser irradiation. Only the laser-treaded group appears to have had a probe inserted into the nasal cavity after anesthesia. For the natural recovery group, insertion of the probe into the nasal cavity and anesthesia must be performed without laser irradiation. Otherwise, it is undeniable that the insertion of the Probe into the nasal cavity, rather than the laser, led to a reduction in inflammation. Nasal procedures affect the nasal mucosa. The improvement seen in the laser treated group may simply be due to increased drainage due to the nasal procedure, not the laser.
The lack of sufficient discussion of the mechanisms by which lasers inhibit inflammation is also problematic.
The authors mention the risk of DNA damage in LLLT, but does this not suggest a possible carcinogenic potential? Sinusitis is a benign disease, and the safety of its treatment must be thoroughly discussed.
Author Response
Point 1: When comparing the laser treated group to the natural recovery group, all procedures must be the same except for the laser irradiation. Only the laser-treated group appears to have had a probe inserted into the nasal cavity after anesthesia. For the natural recovery group, insertion of the probe into the nasal cavity and anesthesia must be performed without laser irradiation. Otherwise, it is undeniable that the insertion of the Probe into the nasal cavity, rather than the laser, led to a reduction in inflammation. Nasal procedures affect the nasal mucosa. The improvement seen in the laser treated group may simply be due to increased drainage due to the nasal procedure, not the laser.
Response 1: Thank you for the valuable advice and we completely agree with the reviewer’s opinion. Theoretically, the same probe should have been inserted into the nasal cavity of the natural recovery group as the sham treatment group to compare with the laser-treated group. We admit to missing this point as you commented methodologically.
However, the laser probe was inserted into the nostril of the rabbits superficially, not deeply because the laser irradiation was performed with the straight-type laser, not the diffusion-type laser as seen in Figure 6A and Additional figure 1. In addition, the size of the probe (4.6 mm width and 4.8 mm height, Additional figure 2.) was not considered large enough to injure the nasal cavities of the experimental rabbits (mean size of rabbit nostril = 7-8 mm, Additional figure 3.). For these reasons, we believe that the insertion of a probe into the nasal cavity has a limited effect on sinusitis. Having taken into consideration your suggestions, we would redesign the experiment to include the sham treatment group but unfortunately due to budgetary constraints and animal ethics, we cannot perform it at this time.
Figure 6. Laser irradiation of rabbits. (A) A low-level laser device was used at the following wavelengths and powers: AlGaInP 670 nm, 3 mW and GaAs 830 nm, 20 mW. (B) Low-Level Laser Therapy was applied to five rabbits (G4) through the nostrils for 20 min for 7 days.
Additional Figure 1. Insertion of laser probe to the rabbit nostril.
Additional Figure 2. Size of laser probe. (4.6mm width and 4.8 mm height)
Additional Figure 3. Size of rabbit nostril. (approximately 7-8 mm)
Point 2: The lack of sufficient discussion of the mechanisms by which lasers inhibit inflammation is also problematic.
Response 2: Thank you for the important comment. We added the discussion regarding possible mechanisms by which lasers inhibit inflammation. There are several experimental reports regarding the anti-inflammatory effect of LLLT, however, there is a paucity of studies regarding its effects on rhinosinusitis. We added the descriptions regarding the anti-inflammatory effect of LLLT, such as the regulation of transcription factors associated with cycylooxygenease-2, reducing neutrophil influx, inhibiting NF-kB, TNF-a and MMP-13 in the discussion section as seen below.
The photobiomodulation effect of LLLT can lead to modulation of cell metabolism and reduction of inflammation.
Pires et. al reported that LLLT can induce an anti-inflammatory effect that regulates the transcription factors associated with cyclooxygenase-2 and through the modulation of the expression of pro-inflammtory micro-ribonucleic acid. Aimbre et. al noted that LLLT reduced neutrophil influx and the expression of IL-1b micro-ribonucleic acid. LLLT can inhibit nuclear factor kappa B (NF-kB), the key inflammatory transcription factor, and the related signaling pathways. LLLT can also suppress another important medicator of inflammation, macrophage-associated inflammatory proteins and pro-inflammatory cytokines. Nambi et. al demonstrated that LLLT can reduce TNF-a and MMP-13 whereas LLLT seems to be ineffective in changing levels of IL-6.
LLLT also reduces the expression of IFN-g, IL-1b, and IL-17.
Point 3: The authors mention the risk of DNA damage in LLLT, but does this not suggest a possible carcinogenic potential? Sinusitis is a benign disease, and the safety of its treatment must be thoroughly discussed.
Response 3: We agree that mentioning the risk of DNA damage in LLLT is inappropriate in the discussion section for the present study. With rhinosinusitis being a benign disease, it is absoultely critical that the treatment does not cause any downstream harm. Rather, LLLT could be used as therapeutic tool for tumors according to laser settings. Hence, we have added the discussion regarding the safety of lasers at these intensities below.
At an experimental level, several studies have reported side effects and DNA injury by LLLT . Hence, there could be a concern regarding carcinogenic potential caused by DNA damage. However, the damage can be reversed. It has been decisively reported that LLLT at therapeutic doses promotes the expression of DNA repair genes and induces changes in the gene expression profile of the irradiated cells. Tam et. al reviewed that the response of cancer cells to LLLT vary and may differ considerably among oncologic types and laser settings.

Reviewer 3 Report
Chronic rhinosinusitis (CRS) represents an important public health problem due to the direct and indirect costs as well as due to the significant decrease in the quality of life of these patients. For this reason, studies that explore new ways of treating CRS are extremely important for ENT practitioners. The present study may mark a paradigm shift in the treatment of CRS. Low-level laser therapy (LLLT) may represent a new therapeutic option in CRS. The current study is extremely well designed: the experimental model devoted to CRS on the rabbit, and the resulst of the LLLT treatment is verified histologically, imaging and by the cytokine profile. The working methods are extremely well documented, the discussions present the potential therapeutic role of LLLT. An excellent study for all those interested in CRS pathology.
Author Response
Point 1: Chronic rhinosinusitis (CRS) represents an important public health problem due to the direct and indirect costs as well as due to the significant decrease in the quality of life of these patients. For this reason, studies that explore new ways of treating CRS are extremely important for ENT practitioners. The present study may mark a paradigm shift in the treatment of CRS. Low-level laser therapy (LLLT) may represent a new therapeutic option in CRS. The current study is extremely well designed: the experimental model devoted to CRS on the rabbit, and the resulst of the LLLT treatment is verified histologically, imaging and by the cytokine profile. The working methods are extremely well documented, the discussions present the potential therapeutic role of LLLT. An excellent study for all those interested in CRS pathology.
Response 1: Thank you for the valuable advice. We believe that the reviewer’s encouragement will help this manuscript a lot. Thank you very much, we really appreciate it.

Round 2
Reviewer 2 Report
The authors have provided answers to the points raised by me. Thank you for your response.
Although some problems with the study design remain, I believe the paper has improved. I have no additional remarks to make.